# The Aging Enteric Nervous System

**DOI:** 10.3390/ijms24119471

**Published:** 2023-05-30

**Authors:** Tinh Thi Nguyen, Peter Baumann, Oliver Tüscher, Sandra Schick, Kristina Endres

**Affiliations:** 1Department of Psychiatry and Psychotherapy, University Medical Center Mainz, 55131 Mainz, Germany; 2Chromatin Regulation Group, Institute of Molecular Biology, 55128 Mainz, Germany; 3Chromosome Dynamics, Telomeres & Aging Group, Faculty of Biology and Institute of Molecular Biology, 55128 Mainz, Germany; 4Institute of Molecular Biology, 55128 Mainz, Germany; 5Leibniz Institute for Resilience Research, 55122 Mainz, Germany

**Keywords:** aging, gut, enteric nervous system, microbiota, gut–brain axis, neurodegenerative disease

## Abstract

The gut and the brain communicate via the nervous system, hormones, microbiota-mediated substances, and the immune system. These intricate interactions have led to the term “gut-brain axis”. Unlike the brain—which is somewhat protected—the gut is exposed to a variety of factors throughout life and, consequently, might be either more vulnerable or better adapted to respond to these challenges. Alterations in gut function are common in the elder population and associated with many human pathologies, including neurodegenerative diseases. Different studies suggest that changes in the nervous system of the gut, the enteric nervous system (ENS), during aging may result in gastrointestinal dysfunction and initiate human pathologies of the brain via its interconnection with the gut. This review aims at summarizing the contribution of normal cellular aging to the age-associated physiological changes of the ENS. Morphological alterations and degeneration of the aging ENS are observed in different animal models and humans, albeit with considerable variability. The aging phenotypes and pathophysiological mechanisms of the aging ENS have highlighted the involvement of enteric neurons in age-related diseases of the central nervous system such as Alzheimer’s or Parkinson’s disease. To further elucidate such mechanisms, the ENS constitutes a promising source of material for diagnosis and therapeutic predictions, as it is more accessible than the brain.

## 1. Introduction

The gastrointestinal (GI) tract is innervated by a vast intrinsic nervous system, the enteric nervous system (ENS), which can function independently of the central nervous system (CNS) [1]. The ENS is the largest and most intricate division of the peripheral nervous system, extending from the upper esophagus to the internal anal sphincter and connecting to the liver, gall bladder, biliary tract, and pancreas [2].

### 1.1. The ENS Structure

The ENS is derived mainly from vagal neural crest progenitors that colonize the gut during fetal development and organize into an interconnected network of nerve and glial cells that are organized into ganglia located in two major plexi: the myenteric plexus and the submucosal plexus [2,3,4]. The two interconnected ganglionated plexi wrap around and integrate into the laminar structure of the GI tract. The myenteric plexus lies between the longitudinal and circular muscle layers and along the entire GI tract. The submucosal plexus lies beneath the mucosa, closer to the intestinal lumen, and in the small and large intestines but not in the stomach or esophagus [2].

The ENS has a bidirectional connection to the CNS via the vagus nerve, pelvic nerves, splanchnic nerves, and sympathetic pathways, including afferent (sensory) innervation and efferent (motor) innervation [5,6]. Sensory information is transmitted both to the CNS via extrinsic primary afferent neurons and to the ENS via intrinsic primary afferent neurons that follow vagal and spinal nerve innervations [5]. Pre-enteric neurons are found in the efferent pathways; they terminate in the enteric ganglia and regulate or alter the activity of enteric neurons. Vagal innervation conveys mechanoreceptive and chemoceptive information from the esophagus, stomach, and intestine to the CNS, but not pain signals, and regulates gastric propulsion and motility, gastric acid production, and hormone release in the ENS [5]. Unlike the vagal nerve, the splanchnic nerves carry pain conducting visceral afferent fibers and visceral efferent fibers that inhibit GI motility and secretion; stimulate gluconeogenesis, glycogenolysis and glucose release, and secretion of catecholamines by chromaffin cells; and allow for transmission of the intractable visceral pain [6].

The ENS comprises a large number of neurons—approximately 200–600 million neurons in the human GI system, that equal the number present in the spinal cord [7,8]. A vast panel of neurotransmitters and neuromodulators is found in the ENS that is similar to those found in the CNS. Single-cell based RNA sequencing approaches have led to the identification of up to 22 enteric neuron subtypes in the mouse myenteric plexus throughout the whole intestine [9] and three glia subsets from the mouse colon [10]. Similarities in gene expression for many enteric neuron subtypes in mice and humans are documented [9,10]. The ENS of mice and humans have a broad overlap, i.e., shared program, which is indicated by the fact that excitatory/inhibitory motor neurons, sensory neurons, interneurons, secretomotor/vasodilator neurons such as expression of calcitonin-related polypeptide beta (*Calcb*), glial-cell-line-derived neurotrophic factor family receptor alpha 2 (*Gfra2*), nitric oxide synthase 1 (*Nos1*), vasoactive intestinal peptide (*Vip*), etc., are found in both species [10]. Moreover, similarities in mice and human can be deduced by the expression of the Y-linked gene (DEAD-Box Helicase 3 Y-Linked, *Ddx3y*) only in males and the expression of X-inactive specific transcript (*Xist*) only in females in the ENS of both [9]. While there are these large similarities described in mice and humans, there were a few differences observed [9,10]. Gene expression differences in the melanocortin, leptin, and serotonin pathways that suppress appetite were observed between human and mouse ENS [10]. While human neurons were reported to be enriched for the melanocortin receptor (*MC1R*), mouse neurons were strongly enriched for the expression of its antagonist (*Agrp*). The leptin receptor (*LEPR*) and the enzyme for serotonin synthesis (*TPH2*, tryptophan hydroxylase 2) were strongly expressed in human enteric neurons but not in mouse neurons. Other genes marking neuronal subtypes were divergent. For example, Neuromedin U *(Nmu*) and Kelch Like Family Member 1 *(Klhl1*) were expressed in murine intrinsic primary afferent neurons (IPANs), whereas *NMU* labeled IPANs and *KLHL1* labeled a distinct population in humans [9].

### 1.2. The ENS Function

The ENS regulates almost all aspects of intestinal physiology such as GI motility, vasodilation, absorption, secretion, and organizing immunoregulatory and inflammatory processes [2]. The myenteric plexus (MP) coordinates muscle movements underlying the propulsion of content, while the submucosal plexus (SP) plays a role in secretion and absorption [11].

At the functional level, enteric neurons can be divided into sensory neurons, interneurons, and motor neurons, but these have multiple functions [3]. For example, sensory neurons are responsive to mucosal mechanical distortion and to chemicals that interact with the mucosa [3,4]. Mechanosensitive enteric neurons throughout the gut respond to compression or stretch rather than to shear stress [3,11]. The chemosensitive enteric neurons possess receptors for amino acids, fatty acids, pH, glucose, tastes, and odorants. Interneurons are involved in the migrating myoelectric complex of the small intestine [4]. Motor neurons regulate epithelial secretion and blood flow and a dysregulation of their activity causes gastrointestinal dysmotility [4].

Furthermore, different functions of the ENS in specific gut regions may be attributed to variations in the amount and gene expression profiles of distinct subsets of enteric neurons [9,10]. For example, multiple glutamate receptors are elevated in the ileum, enabling the efficient absorption of glutamate. On the other hand, the colon may require secretomotor/vasodilator neurons to regulate fluid balance.

Here, we aim to review the current knowledge of age-related changes in the ENS in terms of function, morphology, neuronal populations, pathophysiological mechanisms, and their link to neurodegenerative disorders across different model systems.

## 2. The ENS during Aging

Aging is characterized by the progressive dysfunction of most tissues and organs, which is associated to the gradual decline in regenerative capacity, cell proliferation, telomere maintenance, and genome stability [12]. Since the ENS coordinates GI physiology, the aging ENS may result in GI dysfunction, and symptoms such as dysphagia, gastrointestinal reflux, constipation, and fecal incontinence increase with advanced age [13,14,15,16].

Since some characteristics are shared between human and mouse, mouse models offer access to understanding the inner workings of the ENS. The information obtained with other models could better enrich our understanding of the ENS during aging within different species.

### 2.1. Morphological Changes of the Aging ENS

Over the lifespan, the size of the GI tract changes in terms of the length and circumference of the intestine and the volume of the different layers of the gut [17]. These changes occur throughout periods of gut maturation and potentially impact the arrangement of the enteric plexi and the nerve fibers’ density in different layers [18]. To allow plasticity of the ENS during periods of gut growth, the intrinsic sensory neurons and the interneurons need to lengthen their processes to maintain contact with their target cells, or the size and number of smooth muscle cells have to increase. Therefore, if the ganglia’s size remains constant, or if the density remains constant, it is possible to predict either fewer or more ganglia per surface unit. These phenomena could then also occur with aging.

Age-related changes in morphology of the MP are reported in rodent models, guinea-pigs, and humans (Figure 1). The size of the myenteric ganglia and ganglionic area in the guinea-pig ileum and mouse stomach, jejunum, and colon were reduced with age, and the myenteric neurons appeared to have less packing density within ganglia [19,20]. In human colonic specimens, the proportion of ganglia that appeared to be uniformly filled with neurons was decreased by 35% with age, while the proportion of ganglia that contained areas with bundles of nerve fibers did not change with age (10 days–92 years, *n* = 168) [21]. The area of the ganglia was larger by 0.027 mm^2^, and the percentage of ganglia containing empty spaces increased approximately threefold in the colon from infants to elderly people (10 days–92 years, *n* = 19). The age-dependent changes were greater in the colon than in the ileum. No gender-specific differences were observed. Similar to this observation, in human duodenum/jejunum samples from older individuals (65–84 years, *n* = 10), ganglion structures were often found with no nerve cell bodies due to loss of neurons, but the surface area of the ganglion structures of the MP per se did not decrease [22,23].

Additionally, the distribution and morphology of enteric neurons were studied during aging (Figure 1). There was a reduction in the proportion of intra-ganglionic neurons but an increase in the population of extra-ganglionic neurons in the MP of aging guinea-pig ileum [19]. The neuronal cell area increased in the human esophagus (>70 years, *n* = 4) but not significantly [24]. In the jejunum of older humans (65–84 years, *n* = 19), a reduction in the number of myenteric neurons was accompanied by a significant increase of 14% in neuron size [22]. The surface of neurons and their nuclei in the oldest groups was larger, but the ratio of nucleus surface to cytoplasm did not differ [22]. Phillips et al. showed an increase in neuronal cell body size of approximately 20% in the rat colon at 24 months (*n* = 12) compared to animals aged 3 months (*n* = 14) [25]. A significant increase in the density of the nerve fibers in the MP of the mouse colon was observed with increasing age (between 3–4 months; *n* = 5 and 12–13 months; *n* = 4 and between 12–13 months; *n* = 4 and 18–19 months; *n* = 4) when corrected for gut growth and stretching [26]. The fibers stained for calbindin and neuronal nitric oxide synthase (nNOS) appeared swollen within the ganglia, within interganglionic nerve fiber bundles, in areas adjacent to the ganglia, and in the plane of the MP between ganglia of 18–19- (*n* = 4) and 24–25-month-old mice (*n* = 5) [26,27]. Swollen dystrophic features of tyrosine hydroxylase (TH)- or calcitonin gene-related peptide (CGRP)-positive axons and terminals were observed in the MP of the jejunum (24-month-old rat, *n* = 12), SP of the colon (16–24-month-old rat, animal number not mentioned) [28,29].

### 2.2. Aging ENS Leads to Age-Related GI Innervation Changes

The research on ENS during aging has mostly focused on the analysis of alterations in total neuronal number and sub-populations in the ganglia of the MP and SP, which was studied in a variety of species including guinea-pigs, rodents, and humans. The research was carried out on age-related changes in the ENS through the GI tract from the esophagus to the anal sphincter. Table 1 presents a summary of the studies conducted over the past 30 years, investigating changes in the quantities of enteric neurons and sub-populations during the aging process. These findings build upon the research conducted by Saffrey in 2013 [17].

Many studies suggest that aging results in a loss of myenteric neuronal density [19,22,23,26,29,30,31,34,41,46,47,51,54] and number [29,32,46,52] (see Table 1). For example, Mandic et al. showed a decrease in the number of myenteric neurons per cm^2^ by 25.93% in the aged group (65–84 years) compared to the youngest group (20–44 years) and by 23.32% in relation to the middle-aged group (45–64 years) [22]. A significant loss of myenteric neuron density occurred in both the proximal and distal colons of rats between the ages of 6 and 27 months, with the most dramatic loss of neurons in the distal colon and a loss of 32% of myenteric neurons by 27 months of age [29]. However, several studies have not accounted for changes in gut dimensions that occur with gut growth [19,30,46]. There was a loss of 30% of HuC/D-positive neurons in the MP of the distal ileum of aged rats when changes in intestinal dimensions were not considered [19]. In contrast, two studies showed no change in myenteric neurons when gut growth was considered [39] or not [48]. Furthermore, a few studies have analyzed changes in the number and density of submucosal neurons during aging [29,46,54]. A significant age-related loss of submucosal neuron density occurred in the colon of rats starting at 12 months of age and continuing in a linear fashion through to 27 months, resulting in a loss of 38% of the neuron density and a loss of 24% of the neuron number [29]. A reduction in submucosal neuron numbers per ganglion was also reported in mouse gut [46], but this could not be found in the human colon [54].

The subpopulations of enteric neurons (both excitatory and inhibitory neurotransmission) that are lost during aging were studied by several groups. There is evidence that age-associated loss of myenteric neurons is specific to the ChAT (choline acetyltransferase)-positive cholinergic subpopulation in mice [20] and in humans [54]. The changes in density of NOS-positive nitrergic neurons varied in human studies, even using the same method of NADPH-d staining [21,50,56]. There was an increase of 36% in nitrergic neurons in a group of 78–86-year-old people compared to 42–71-year-old people [50], while a non-significant increase in the proportion of NADPH-d per ganglion with age was found in the MP of the ileum and colon [21]. In contrast, the gene expression levels of *NOS1* decreased and *ChAT* increased with age in the MP of the human colon [56]. The NOS-positive cell numbers in the MP of the colon were reduced by 16.6% in aged donors (70–95 years) compared to babies [56]. Several animal studies reported no change in nitrergic neurons [25,26,31,32], but some others observed a significant decrease [20,32,38,39,43,44] or increase [35,42] (see Table 1). Similar observations were made for substance P-positive neurons. Their density indicated no change in mouse intestine [47], a decrease with age in the mouse internal anal sphincter [48], but an increase in the ileocecal sphincter of aged rats [45]. Some studies reported no change in calbindin-positive neurons in rodents [26,30], but Thrasivoulou et al. showed a reduction with age [36]. In guinea-pigs and rats, an age-related reduction in the density of calretinin-positive myenteric neurons was found with age [19,30,36]. Others observed slight decreases in the number of neurocalcin α-immunoreactivity (IR) enteric neurons in aged rats [40]. The changes in these calcium-binding proteins may indicate alteration and/or maintenance of calcium homeostasis in enteric neurons during aging. The density of another inhibitory neuronal marker, VIP, was found to decrease with age in the rodent sphincter [45,48]. Interestingly, CGRP-positive inhibitory neurons were increased in the sphincter of the aged mice [48] but decreased in that of the aged rat [45]. These findings suggest that neurons expressing calcium-binding proteins might be more sensitive to the aging process than other neuronal subpopulations.

There is only one study that tested γ-aminobutyric acid type A receptors (GABAAR) regarding gene expression level in human colon muscle strips [55]. It reported a decrease in the *GABAARα3* subunit and a slight increase in the α2 and γ2 subunit but without reaching statistical significance in the old group (66–93 years). Another study in humans revealed no change in cellular senescence or oxidative stress (*SNCA*, α-synuclein; *CASP3,* caspase 3; *CAT,* catalase; *SOD2,* superoxide dismutase 2; and *TERT,* telomerase reverse transcriptase) but a decrease in gene expression encoding sodium channel Nav1.1 and 1.5 during aging, which may cause alterations in neurotransmission and gut function [56].

Not only changes in enteric neuronal population but also in enteric glial cells (EGCs) were reported in a few studies. Ginneken et al. observed no change in GFAP-positive EGCs but an increase in S100-IR glial cells in mouse intestine (18 months) [47]. In contrast, significant reductions in numbers of S100-IR glial cells were observed in the rat intestine (26 months) [41]. In another study on the mouse colon, dysregulation in EGCs expression of connexin-43 mRNA and protein within the MP were associated with altered motility with increasing age (12–15 months) [49]. Moreover, S100-IR EGCs in the human descending colon decreased in both the MP and SP from patients (24–78 years) with slow transit constipation [53]. Baidoo et al. reported that the human descending colon during aging (66–81 years) did not display a loss of SOX-10-IR EGCs in the MP and SP but reduced S100-IR EGCs density within circular muscle and MP and little or no GFAP-IR EGCs in both the adult and elderly colon [57]. This alteration in myenteric glial density with age may contribute to colonic dysfunction.

The published data on age-associated changes in the ENS differ widely, and there are various reasons why the evidence is contradictory. Among them are, for example, the use of different species, sampling methods, choice of neuronal markers, difference in gut growth and arrangement of the plexi among species, diet, and environmental enrichment [17,31]. Therefore, further investigation with a careful consideration of these parameters will provide valuable information about the nature and possible causes of neuronal aging in the gut. Additionally, it is rather difficult to compare the age of humans and animal models, even if there are attempts to correlate them (e.g., Dutta and Sengupta [58]).

## 3. Pathophysiology of the Aging ENS

Increased age is associated with many changes in the ENS that impact gut function, as mentioned above. This section focuses on pathophysiological mechanisms and how various factors contribute to the aging ENS, which are summarized in Figure 2.

### 3.1. Aging Phenotypes of the ENS

Enteric neurons in aging mice were reported to exhibit senescence-associated phenotypes, such as DNA damage foci incorporating activated histones such as cH2A.X, elevated levels of reactive oxygen species (ROS), and activated p38 mitogen-activated protein kinase, and to display elevated levels of interleukin-6, senescence-associated β-galactosidase activity, and aggregated proteins (α-synuclein and hyperphosphorylated tau) [59,60,61].

Advanced age is associated with a reduced DNA repair capacity, resulting in an accumulation of unrepaired DNA damage in neurons. This results in enhanced inflammatory mediators, the production of ROS, and mitochondrial dysfunction at the cellular level [12,62,63]. For example, ROS levels were higher in myenteric neurons in the ileum of aged rats compared to those of young animals [60,64]. Because of their large size, high rate of metabolic activity, and weak antioxidant defense, neurons appear to be particularly vulnerable to damage by free radicals [36]. Decreased neuronal mitochondrial quality and activity impair mitophagy, which subsequently triggers protein accumulation via increased oxidative damage and cellular energy deficit, resulting in synaptic dysfunction or even neuronal death.

Calcium dysregulation is considered a factor that additionally contributes to neuronal aging [17]. There is reduced efficiency in calcium absorption with age [65]. The elderly fail to increase calcium absorption when facing a low-calcium diet, whereas younger people have an adaptive response. A decrease in the number of enteric neurons expressing calcium-binding proteins such as calbindin, calretinin, and neurocalcin in aging animals (Table 1) could affect these proteins’ overall levels, which is indicative of calcium dysregulation. The presence of α-synuclein aggregate immune-positive calbindin- and calretinin-IR myenteric neurons during aging [61] suggests that these neuronal subtypes may, in fact, be vulnerable to age-associated degeneration [17,40]. Thus, while decreased mitophagy, calcium-dyshomeostasis, and DNA damage are generally known as hallmarks of cellular aging, they might affect neurons and, with this, ENS more prominently.

### 3.2. Changes of Neurogenesis of the ENS with Aging

The ENS is derived from self-renewing enteric neural crest-derived cells (ENCC) [18,66], vagal Schwann cell precursors [67], and endoderm-derived pancreatic duodenal homeobox-1 (Pdx1)-Cre lineage progenitor cells [68] in the early post-natal stage.

The previous studies suggested that in the adult ENS, ENCC are present [69] and that enteric glia, additionally, have the potential to generate neurons in vitro [70,71,72]. However, reports have apparently failed to find evidence of ongoing neurogenesis in adulthood by following the fate of adult Sox10-expressing cells, which were assumed to be adult enteric neural precursor cell (ENPC), except in in vitro culture or under injury/stressed states. Kulkarni et al., for example, demonstrated, by using BrdU labelling, in vivo enteric neurogenesis of adult ENPC that do not express Sox10 but do express Nestin (2–6-month-old mice) [73]. The adult ENPC not only replaced the neurons born during early development with adult (Sox10−) neurons but also maintained adult neuronal populations by generating neurons at a steady state to substitute for neuronal loss because of apoptosis.

During ENS maturation, a new population of mesoderm-derived enteric neurons (MENs) arrives and continuously expands by hepatocyte growth factor (HGF) signaling [74]. Its presence results in an equal proportion of all myenteric neurons in the ENS in young adulthood (2-month-old mice), reaches 75% of all myenteric neurons by 6 months old and populates the ENS almost exclusively in the aged small intestine (17-month-old mice). These eventually outnumber the neural crest-derived enteric neurons (NENs), which relates to downregulation of glial derived neurotrophic factor (GDNF) signaling through its receptor RET in the adult ENS [74]. Due to increasing proportions of MENs, the dominant population in the aging ENS are CGRP-, P-Cadherin 3-, and MET- (receptor for HGF-positive neurons) expressing neurons. Similarly, slow tonic myosin heavy chain protein- and MET-expressing enteric neurons were also found in the gut of adult humans, suggesting that the mesoderm-derived ENS might be a shared feature between mouse and human. A shift in the balance between the two lineages—NENs and MENs—in the adult ENS may be associated with the intestinal dysmotility observed in aging humans/mice [74,75].

### 3.3. Effect of Dysbiosis on the Aging ENS

Elderly populations exhibit dysbiosis, decreased microbial diversity, and increased representation of proteolytic bacteria, which affect microbial-derived metabolites [76]. The intestinal microbiota is a key regulator in the development of the ENS, its maturation, and neural protection [77,78,79,80,81,82]. It was reported that the number of enteric subpopulations is microbiota-dependent. There was, for example, a decrease in the myenteric neuronal number and nerve density but an increase in the proportion of myenteric nitrergic neurons in the ileum as well as the jejunum in germ-free (GF) mice compared to conventionally raised (Conv-R; normal undefined intestinal microbiota) mice [77]. As a consequence, the amplitude and the frequency of muscle contractions decreased in the ileum and jejunum in GF mice compared to the Conv-R groups. The highest number of PGP9.5 Nestin-GFP double-positive neuron cells was seen in the caecum and proximal colon, which may represent the part of the intestine with the highest microbiota-driven impact on neurogenesis due to the high abundancy and diversity of microbes [78,79]. The longitudinal muscle/myenteric nerve density was reduced in antibiotic-treated mice compared to controls [81]. Moreover, βIII-tubulin-positive innervations of colonic crypts were reduced in GF mice and in antibiotic-treated mice but were restored by microbiota inoculation after 15 days (Conv-D mice; GF mice colonized with the microbiota from Conv-R aged-matched donor mouse’s cecum). Antibiotics mix (Abx)-treated mice exhibited loss of enteric neurons in the ileum and proximal colon in both SP and MP and a reduction in the number of enteric glia in the ileal MP [82]. These observations were accompanied by alterations in GI structure and function. Recovery of the microbiota restored intestinal function and stimulated enteric neurogenesis, leading to increased numbers of enteric glia and neurons. The administration of lipopolysaccharide enhanced neuronal survival in Abx-induced bacterial depletion but showed no effect on neuronal recovery after neuronal loss was mediated by Abx-treatment. In contrast, short-chain fatty acids (SCFAs) were able to restore neuronal numbers even after Abx-induced neuronal loss, suggesting that these bacterial metabolites stimulate enteric neurogenesis in vivo. Moreover, the migration of EGCs toward the submucosal layer is dependent on stimulation by the microbiota [80]. The density of mucosal EGCs decreased significantly in GF mice but was restored upon recolonization [80].

The underlying mechanism of microbiota-driven ENS maturation and aging was discussed elsewhere [81,83,84,85,86,87,88]. The serotonin (5-hydroxytryptamine, 5-HT, a neurotransmitter or hormone that modulates intestinal secretion and motility) and 5-HT receptor 4 expressions in the gut are both microbiota-dependent, which is shown by their absence in GF mice, restored expression in Conv-D mice, and presence in Conv-R mice [81]. By secreting the enzyme β-glucuronidase, which converts estrogens into their active forms, intestinal bacteria modulate the estrogen signaling capacity [84]. This process may be disturbed by intestinal microbiota dysbiosis, accompanied by lower microbial diversity, as the level of circulating estrogens is then reduced. It is known that estrogen regulates 5-HT synthesis, reuptake, and degradation [89,90] and contributes to enteric neurogenesis in vitro and in vivo [91]. Therefore, the microbiota could affect the estrogen level, which, in turn, might influence 5-HT level, eventually leading to alterations in neurogenesis (Figure 3). Aged mice and macaques exhibited a decrease in beneficial gut bacteria, such as *Akkermansia muciniphila* and SCFA-producers in *Clostridium* members of cluster IV, and an increase in pro-inflammatory microbes, which may eventually lead to aging-related ENS degeneration because of gut dysbiosis [85]. *C. difficile* toxin A application, for example, increased the activity of both S- (synaptic or Dogiel Types I) and AH (after-hyperpolarization or Dogiel Types II)-type submucosal neurons in guinea-pigs [86]. Furthermore, the interaction between gut microbiota and EGCs relies on Toll-like receptors (TLR) 2 and 4, which are expressed in EGCs [80]. The expression of GDNF and GFAP-, S100β-positive glia was markedly decreased in the MP of TLR2^-/-^ mice, while the administration of GDNF rescued the ENS deficit [80]. Taken altogether, modifications in diet (prebiotics) or the introduction of particular bacteria could alter enteric neurogenesis and thereby improve gut physiology in ENS-related disorders or aging-related functional decline.

### 3.4. Interactions between Different Cell Types and the ENS in Aging

#### 3.4.1. Enteric Immune System

Neuroimmune interactions were reported between muscularis macrophages (MMs) and the ENS in the MP [87,92,94] (see Figure 3). MMs express high levels of β2 adrenergic receptors, which is essential for noradrenergic signaling, and MMs reside close to the calcium indicator GCaMP3-labelled enteric neurons [94]. During homeostasis, enteric neurons support the survival and development of MMs by their secretion of colony stimulatory factor 1 as a growth factor [87,92]. In turn, bone morphogenic protein 2 (BMP2), which is expressed by MMs, stimulates enteric neurons expressing BMP2 receptors to control colonic peristaltic reflex. Moreover, MMs in the postoperative ileum of mice produce lower pro-inflammatory cytokines in response to acetylcholine released from ChAT-positive myenteric neurons by improving postoperative gastric emptying [92]. MMs may also contribute to ENS homeostasis by phagocytosis of apoptotic neurons, thereby supporting neuronal turnover. Interestingly, a shift in intestinal macrophages from an anti-inflammatory M2 phenotype to the pro-inflammatory M1 phenotype during aging leads to intestinal inflammation and is accompanied with increased apoptosis and loss of enteric neurons and ENPCs and intestinal transit. Additionally, aging-associated enteric neuronal degeneration may also be inflammation independent. For example, acetylcholine-expressing neurons are documented to be selectively vulnerable to age-related loss, and acetylcholine, acting via the α7-subunit of the nicotinic acetylcholine receptor, can convert M1 macrophages to the M2 phenotype. Hence, it is necessary to understand the close interplay between MMs and their surrounding kinds of enteric axons.

#### 3.4.2. Enteroendocrine Cells

The most dominant subtype of intestinal enteroendocrine cells (EECs) in the colon is the enterochromaffin cell type (ECC). These cells communicate with luminal bacteria and interact with the afferent and efferent nerve terminals in the lamina propria by activated chemosensory receptors [95]. Though ECC comprises only 1% of intestinal epithelial cells, they produce and release 90% of the body’s 5-HT [96,97]. Chen et al. demonstrated that under environmental stress, epithelial cells released interleukin-33, which binds to suppression of tumorigenicity 2 receptors on ECC. This triggers the release of 5-HT and, subsequently, modulates enteric neuronal activities and increases colonic motility [98]. The enteroendocrine L cell subtype produces and secretes glucagon-like peptide-1 (GLP1), GLP2, peptide YY (PYY), and oxyntomodulin. GLP1 and GLP2 have neuroprotective effects and trigger neurogenesis as well as anti-inflammatory effects [95], whereas PYY has an anti-secretory effect and prolongs intestinal transit [93]. Although some neuroendocrine molecules secreted by EECs have a beneficial effect, they are observed to correlate with multiple neurological and psychiatric disorders [95]. The number of EECs was reported to increase with age in mouse, but their 5-HT secretory activity was decreased [99]. In contrast, the increase in number of PYY-positive cells in old mice accounted for the increase in the secretion of PYY [93], which could contribute to the delayed colonic transit, mediated by Y2 receptors on the ENS and the development of constipation at advanced age (Figure 3) [93].

#### 3.4.3. Tuft Cells

Tuft cells are chemosensory cells located in the intestinal mucosa that decrease in density from the upper to the lower GI tract and play an essential role in mucosal maintenance and immunity in response to multiple stressors [100]. The studies observed an anatomic connection between epithelial tuft cells and enteric ganglia in the lamina propria emanating from the submucosa within the GI tract [100,101]. This suggests the potential of a bidirectional communication with the ENS (Figure 3). Tuft cells and submucosal and myenteric neurons express the SCFA receptor (FFAR3, free fatty acid receptor 3), the stimulation of which may modulate neuronal activity [97]. Both tuft cells and myenteric excitatory neurons express ChAT [97], and reduced ChAT signaling from enteric neurons appears to provoke an increase in tuft cells to maintain cholinergic homeostasis [102]. Tuft cells were reported to lack both vesicular acetylcholine transporters and high-affinity choline transporters, which are needed for the synaptic release of acetylcholine [103]. Although more research is needed to determine the mechanism for putative acetylcholine secretion from tuft cells, taken together, these observations highlight the importance of tuft cells in neuronal-epithelial cross talk.

### 3.5. Neurotrophic Factors as Protectors of Enteric Neurons along Aging

Neurotrophic factors are essential to neuronal survival and differentiation during development and contribute to the antioxidant defense of neurons during the life span [36]. The NEN population is regulated by GDNF signaling through its receptor RET [74]. GDNF and neurotrophin-3 (NT-3) were found to reduce ROS levels in myenteric neurons in caloric restriction- and ad-libitum-fed rats at the age of 12–15 months, but this effect was only observed in caloric-restriction-fed aged animals (24 months) [36]. GDNF had a protective role against menadione-induced myenteric neuronal cell death in intact ex vivo gut preparations [36]. NT-3 protected enteric neurons against hydrogen-peroxide-induced neuronal cell death in myenteric ganglia culture in vitro [104]. Given their significant role, the disruption of neurotrophic factor support was suggested as a mechanism contributing to enteric neuronal aging [16].

### 3.6. Modulation of Enteric Populations by Dietary Supplements

People change their eating behavior when they get older [105], as do experimental animal models [106,107]. It probably has an impact on the ENS, subsequently raising digestive problems with age [13]. Several studies have shown an effect of dietary supplements on the ENS. Supplementation with certain nutrients, such as omega-3 fatty acids, can attenuate neuronal loss with aging in rodent models [76]. The feeding of rats with a resistant-starch-supplemented diet, which induced the production of the SCFA butyrate, selectively increased the number of cholinergic neurons in the colon [108]. The administration of ascorbic acid as an antioxidant had a neurotrophic effect on the ileum VIP-ergic neurons of Wistar rats after 120 days of supplementation [108,109], while no neuroprotective effect on myosin-V-IR myenteric neurons was observed [110]. The administration of sialic acid resulted in differential expression of the colonic neuronal markers *nNOS* and ubiquitin carboxyterminal hydrolase L1 *(Uchl1*) in young and aged rats, indicating that regulatory mechanisms change with age [111]. There is also increasing evidence that persistent exposure to a high-fat diet (HFD) can result in ENS sensitization and loss and/or damage of enteric populations and gut dysmotility [112,113]. In mice, the HFD caused obesity, hyperglycemia, and insulin resistance after four weeks of feeding and was accompanied by a significant decline in the area density of mucosal S100β-positive glial cell networks at 8 and 20 weeks, while myenteric glial cells were unaffected by early and late feeding periods [113]. However, *Sox10* transcript levels and immunoreactivity revealed a diet-independent, age-associated reduction in glial cell populations. In another study, where myenteric neuron cultures were treated with palmitic acid (a free fatty acid constituent of several HFD), neuronal death and myenteric neuronal dysfunction were induced, including chromatin condensation, decreased acetylcholine synthesis, membrane deterioration, cell shrinkage, and increased oxidative stress [112]. Palmitic acid did not alter the density of cultured EGCs but induced morphological changes from a stellate to rounded phenotype. Therefore, HFD-feeding might affect EGC function since this shifted phenotype is a characteristic of reactive glia.

The previous research in rats suggests that caloric restriction prevents neuronal loss [35,36,38,114,115]. For example, a 50% reduction in ileal myenteric neurons from ad-libitum-fed rats (24-month-old) was prevented by a 30% caloric restriction [38]. NOS-IR neurons did not show diet-related changes in the ileum, but NOS protein and release of nitric oxide decreased in the colonic MP in the aged rats. Caloric restriction might protect neurons from age-related cell death by increasing the antioxidant defense mediated by the neurotrophic factor [64].

## 4. Potential Disease-Related Pathology of the ENS

The human ENS system changes gradually during aging, which is often accompanied by the occurrence of neurodegenerative disorders [87,88,116]. A high overlap between the CNS and ENS, such that neurons use the same kind of neurotransmitters and are supported by glial cells in both systems, suggests that a disease process affecting the CNS could involve or be mirrored by enteric neuronal dysfunction [117]. Additionally, GI dysfunction symptoms such as chronic constipation and fecal incontinence are frequently associated with late stages of dementia and Parkinson’s disease. Therefore, the aim of the current work was to provide information on a potential mechanism by which changes in the aging ENS may contribute to the onset and progression of neurodegenerative diseases such as Parkinson’s disease or Alzheimer’s disease.

### 4.1. Involvement of the ENS in Parkinson’s Disease

Parkinson’s disease (PD) is characterized by abnormal alpha-synuclein (α-Syn) neuronal aggregates, known as Lewy bodies/Lewy neurites, and the selective degeneration of midbrain dopaminergic neurons, which results in motor symptoms [118]. Environmental toxins and gut dysbiosis may trigger oxidative stress and mucosal inflammation, which initiate α-Syn accumulation in the ENS early in PD [87,88]. Enteric neurons in the aged murine and human colon were found expressing risk genes for Hirschsprung’s disease (*RET*, *PHOX2B*, *GFRA1*, and *ECE1*), autism spectrum disorders (*NRXN1*, *ANK2*, *DSCAM*, and *GABRB3*), inflammatory bowel disease (*BTBD8*, *GRP*, *CNTNAP2*, etc.), and PD (*DLG2, SNCA*, *Lrrk2, Park2*, etc.), suggesting ENS contribution to these diseases [9,10,119]. Lewy bodies were found in enteric neurons of PD patients, e.g., within the SP of the colon where dopaminergic neurons are located [120], even preceding the development of motor defects by several years [87,121]. Additionally, these proteinaceous deposits were found in nNOS-, calbindin-, and calretinin-IR enteric neurons of Fischer 344 rats at the age of 18 and 24–25 months [61]. It was suggested that the ENS is an initial site of α-Syn aggregation and that the aggregates subsequently spread to the CNS through vagus nerve fibers. Indeed, patients and mice undergoing vagotomy exhibit a reduced risk of developing PD [116,122,123]. The gut injection of α-Syn fibrils at the MP of the duodenum and pylorus (3-month-old mice) converted endogenous α-Syn to a pathologic species that spread to the brain via the vagus nerve and caused PD-like motor and non-motor symptoms [116]. The inoculation of α-Syn fibrils at the MP of the duodenum in aged mice (16 months old), but not younger mice (2–3 months old), resulted in the progression of pathological α-Syn to the midbrain and subsequently elicited motor deficits [124]. Moreover, upon gastrointestinal injection of aggregated α-Syn, a robust age-dependent gut-to-brain propagation of α-Syn pathology along the sympathetic and parasympathetic nerves caused age-dependent dysfunction of the heart and stomach in rats [125], as seen in PD patients [120,126].

### 4.2. Involvement of the ENS in Alzheimer’s Disease

Alzheimer’s disease (AD) is characterized by the accumulation of Aβ peptides into plaques and hyperphosphorylated tau tangles in cortical regions and associated with cognitive decline [127]. Similar to PD, AD-related pathologic aggregates were also observed in the ENS in few studies [128,129,130]. In Sprague-Dawley rats, a high presence of hyperphosphorylated tau was found in cholinergic neurons and an increased percentage of hyperphosphorylated tau was expressed in the ileal nitrergic subpopulation with an adaption to very low core body temperatures [129]. An AD mouse model, AβPP/PS1 (13 months old), demonstrated robust neuronal Aβ-IR in both the ileal MP and SP [131]. Additionally, both Aβ and phosphorylated tau protein were observed in the ENS neurons of the colon of AD patients. In contrast, Yelleswarapu et al. could not detect Aβ accumulation in the colonic MP of 5xFAD mice and GI dysmotility at 6 months of age [132]. However, while using the 5xFAD mice, our own previous studies showed Aβ depositions in the muscular layer, as well as the MP and SP of the duodenum and ascending colon [130,133]. The 5xFAD mice additionally exhibited an enlarged mesh width of colonic MP and higher incidence of elongated primary cilia in a primary enteric neuron culture, together with altered gut architecture and function (6-month-old mice) [134]. This pointed out that disturbed intercellular communication of the ENS is linked to AD pathogenesis. Moreover, the injection of Aβ, tau fibrils, or AD patient brain extracts into the colon of 3xTg AD mice triggered AD pathology transmission from the gut to the brain via the vagus nerve and caused cognitive dysfunction [135].

Overall, although still debated, the ENS may be involved in the development of CNS diseases by triggering pathological aggregates and transmitting them to the CNS via neuronal connections.

## 5. Conclusions

Over the course of a lifetime, the ENS continues to change. There is evidence that aging is associated with enteric neurodegeneration, which we discussed in this review, but these changes are also attributable to a non-pathological plasticity of the system, which is unquestionably favorable. It is now increasingly clear that diet and changes in microbiota, together with intestinal inflammation and different cell type interactions may influence aging of the ENS, subsequently resulting in age-related GI dysfunction. Understanding the different contributions of cellular aging, the microbiota, nutrition, lifestyle, comorbidities, and medicine to ENS aging is, therefore, important to develop preventative and therapeutic approaches.

Considering the nature of the field, it is important that more studies should be conducted to interpret how the ENS changes during aging, and especially during diseased aging in terms of molecular signatures, and how these changes affect the gut function and its interaction with the aging brain, e.g., by using intestinal organoid models. A better understanding of this concept would open up new diagnostic approaches (gut neurons as a proxy for the CNS), allow for therapeutic testing, and offer new treatment options for those suffering from neurodegenerative diseases.

## Figures and Tables

**Figure 1 ijms-24-09471-f001:**
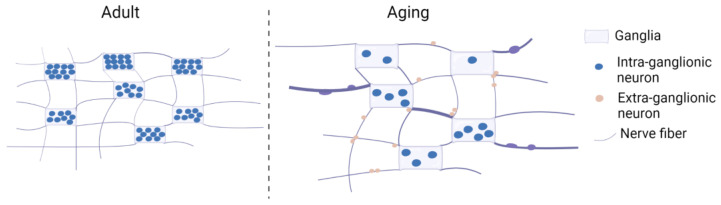
Morphological changes in the aging ENS. During aging, enteric neurons appear to have less packing density within ganglia and are accompanied by an increase in neuron size. The area of the ganglia is larger, and the proportion of ganglia containing empty spaces increases. The proportion of intra-ganglionic neurons is reduced, whereas the population of extra-ganglionic neurons increases. Dystrophic features of swollen axons and terminals are observed.

**Figure 2 ijms-24-09471-f002:**
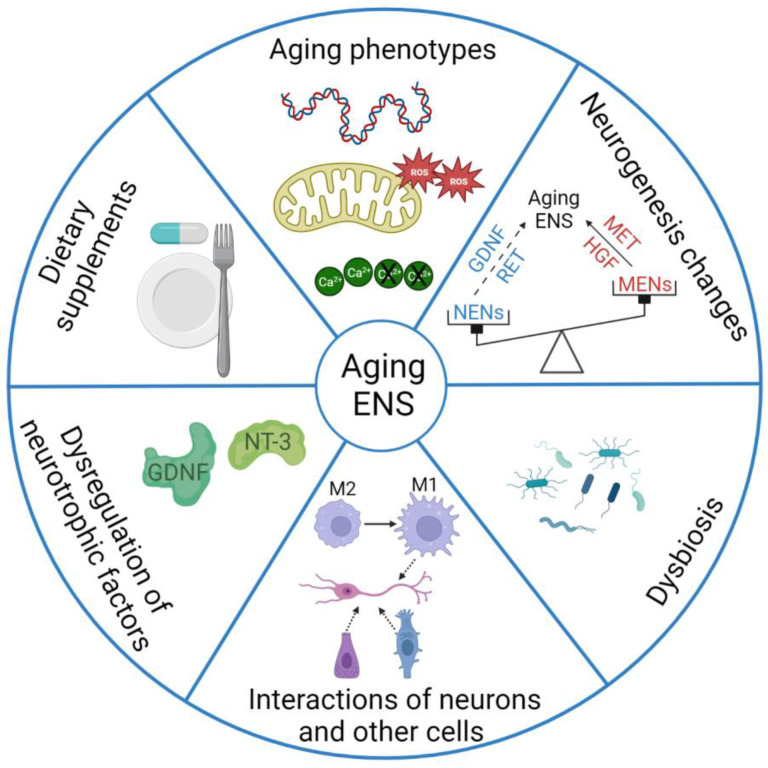
Pathophysiological mechanisms of the aging ENS. The scheme enumerates the six hallmarks that directly affect the ENS during aging. Those include aging phenotypes of the ENS, changes in neurogenesis of the ENS, the effect of dysbiosis on the aging ENS, interactions between different cell types and the ENS in aging, neurotrophic factors as protectors of enteric neurons during aging, and modulation of enteric populations by dietary supplements. Abbreviations: GDNF, glial derived neurotrophic factor; NT-3, neurotrophin-3; NENs, neural crest-derived enteric neurons; HGF, hepatocyte growth factor; MET, MET proto-oncogene or hepatocyte growth factor receptor; MENs, mesoderm-derived enteric neurons; M1, pro-inflammatory M1 macrophage; M2, anti-inflammatory M2 macrophage.

**Figure 3 ijms-24-09471-f003:**
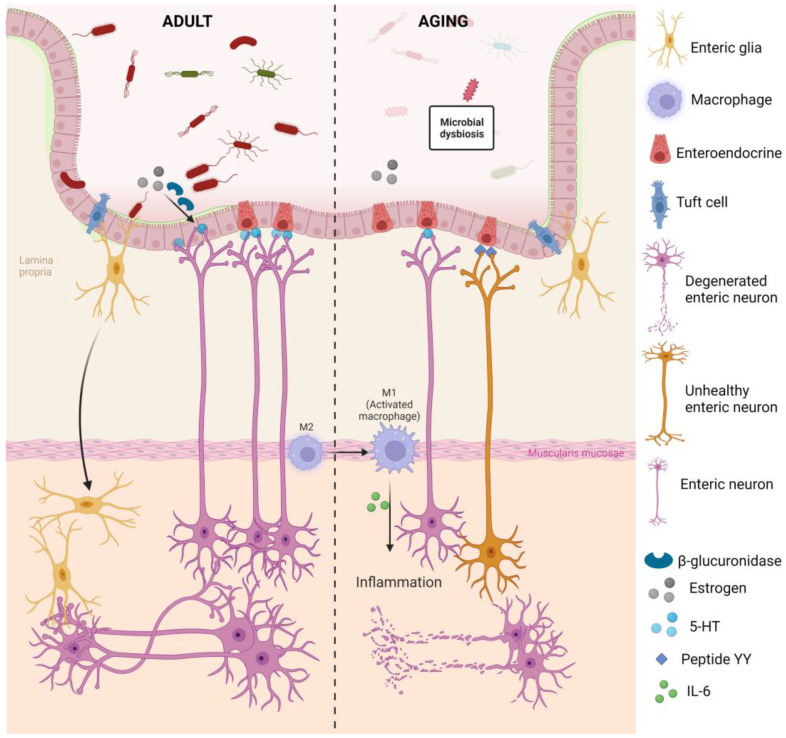
Interactions of different cell types and microbiota within the ENS during aging. Microbiota release β-glucuronidase to regulate estrogen signaling capacity, thus, regulating 5-HT synthesis. This 5-HT is accompanied by that synthesized by enteroendocrine cells modulating enteric neurogenesis. During aging, dysbiosis and enteroendocrine cells with low 5-HT secretory activities decrease neurogenesis capacity. A macrophage shift to the activated M1 phenotype leads to intestinal inflammation with the secretion of pro-inflammatory cytokines, eventually eliciting neuronal degeneration [92]. In addition, increased PYY secretions with age contribute to dysfunctional neurons [93].

**Table 1 ijms-24-09471-t001:** Change in numbers of enteric neurons and ENS sub-populations during aging.

Species and Strains	Regions	Age	Parameters for Identification of ENS Populations	Changes	Ref.
**Guinea-pig**					
♀	Distal ileum (MP)	1–3 m (*n* = 7), 8–10 m (*n* = 5), 22–26 m (*n* = 5)	Total (HuC/D), calretinin/ganglionic area and /area	~30% loss of total neurons ~50% loss calretinin-IR neurons	[19]
Dunkin-Hartley ♀	Distal ileum (MP)	2–3, 8–10, 22–26 m (*n* = 4–6 per age)	Total (HuC/D), nNOS, calbindin, calretinin, neurofilament/ganglionic area	Total neurons: decreased in 22–26 m, not changed in 8–10 m. Calbindin, calretinin, neurofilament alone: no change at 22–26 m. Co-localized calretinin and neurofilament and NOS alone: loss in 22–26 m.	[30]
Tri-color strain	Mid-colon (MP)	2 weeks (*n* = 7), 6–8 m (*n* = 30), 24–32 m (*n* = 30)	Total (HuC/D), nerve fibers (nNOS and substance P)/area and ganglionic area, nNOS and substance P	Significant loss of 56% neurons between 6 m and 24 m but 19% non-significant loss after corrected. Significant reductions in percentage area for NOS- and substance P-IR nerve fibers in circular muscle but no change after corrected. Increase in percentage area of NOS-and substance P-IR between 6 m and 24 m. (Corrected for increasing area of the colon wall)	[31]
**Rat**					
Sprague-Dawley and Wistar	Esophagus (MP)	3–4.5 m (*n* = 10), 18–20 m (*n* = 8)	Total (PGP 9.5), nitrergic (NADPH-d and nNOS)/ganglion	Total neuronal loss ~27% in both strains. Nitrergic neuron reduction in Sprague-Dawley, not in Wistar.	[32]
Fischer 344 ♂	Stomach, small intestine, colon, rectum (MP)	3, 12, 21, 24, 27 m (*n* = 108)	Total (Cuprolinic Blue)/area	No loss in antrum and corpus at 24 m, 38% loss in fore stomach at 27 m; loss of 17–31% in small intestine, 38–39% in colon and rectum at 27 m. (Correction for differences in intestinal size)	[33]
Fischer 344 ♂	Stomach, duodenum, jejunum, ileum, colon, rectum (MP)	3 and 24 m (*n* = 40)	Total (Cuprolinic Blue), nitrergic (NADPH-d)/area	Total: 3–7% loss in stomach, 10–27% loss in small intestine regions, 37–41% loss in colon and rectum. No age-related changes in the density of NADPH-d-labeled neurons in all regions. (Correction for differences in the circumference of the tissue)	[25]
Wistar ♂	Duodenum (MP)	21, 60, 90, 210, 345, 428 days (*n* = 60)	Total (Geimsa and myosin V-IR)/area	Reduction in neuronal density and number over time. (No correction for changes in intestinal dimensions)	[34]
Wistar ♂	Duodenum (MP)	6, 18 m (*n* = 30)	Total (Geimsa), nitrergic (NADPH-d)/area	29% total loss, an increase of 20.4% in density of nitrergic neurons. (No correction factors applied because intestinal area did not differ with age)	[35]
Sprague-Dawley ♂	Ileum (MP)	6, 12, 13, 17, 24 m (*n* = 105)	Total (PGP9.5, HuC/D), calretinin, calbindin/area	PGP9.5 (24 m): 51% loss, HuC/D (17 m): 50% loss, calretinin (17 m): non-significant loss (~30%), calbindin (17 m): 62% loss.	[36]
Wistar ♂	Ileum (MP)	4, 24, 30 m (*n* = 5 each group)	Nitrergic (NADPH-d)	15% loss at 24 m and 30 m (Length of the intestine was considered)	[37]
Sprague-Dawley ♂	Ileum (MP)	4–6 (*n* = 16), 16 (*n* = 4), 24 m (*n* = 9)	Total (PGP 9.5), ChAT-IR, NADPH-d/area	52% total loss of total at 24 m, 64% loss of ChAT-IR neurons, 16% loss of nitrergic neurons at 24 m. (Correction factors of length and circumference of the ileum)	[38]
Sprague-Dawley	Small intestine (MP)	4 m (*n* = 8) and 24 m (*n* = 9)	PGP 9.5, NADPH-d, ChAT, and VIP	Reduction by 15% number of nitrergic neurons in old rats. No reduction in PGP 9.5-IR neurons. No significant reductions with age in other neuronal markers. (Correction for growth of the intestine)	[39]
Wistar ♂	Esophagus to distal colon (MP, SP)	3 m (*n* = 6) and 24 m (*n* = 6)	Neurocalcin-α-IR	No changes in the esophagus and stomach, increase in the pylorus and slight decreases in the small intestine and colon, no decrease in the distal colon.	[40]
Fischer 344 ♂	Duodenum, jejunum, ileum, colon, rectum (MP)	5–6 m (*n* = 8), 26 m (*n* = 8)	Total (HuC/D) neurons/ganglionic area, glia S100/ganglionic area, and/neuron	Significant reductions in numbers of glia and neurons in all regions of aged rats, but non-significant decrease in rectum. (Normalizing with the “dilution” effects of experimental stretch)	[41]
Wistar ♂	Ileum, proximal colon (MP)	E-d19, P-d4, 6 m, 26 m (*n* = 6 per age)	Nitrergic (NADPH-d), PGP 9.5/ganglion	Increased proportion of nitrergic neurons per PGP9.5-IR neurons in colon, but not in ileum at 26 m.	[42]
Specific pathogen-free Fischer 344 ♂	Proximal colon (MP, SP)	6, 31, 74 weeks, 2 y (*n* = 5 per age)	nNOS-IR, PGP 9.5-IR	Significant decline in *nNOS* mRNA, protein level, and nNOS-IR nerve fiber with age. Decrease in the relative ratio of nNOS/PGP 9.5-IR and the percent of nNOS-IR neurons. (Exclude the dilution effect by growth)	[43]
Fisher (F344XBN)F1	Mid-colon (MP)	4–8 m (*n* = 24), 22–28 m (*n* = 24)	PGP 9.5 protein levels, NOS protein and mRNA, nNOS/ganglion	No change in PGP9.5 levels. Significant reduction in number of nNOS-IR neurons per ganglion. Reduction in NOS protein (54 ± 14%) and mRNA (35 ± 15%).	[44]
Fischer 344 ♂	Proximal and distal colon (MP, SP)	6, 12, 18, 24, 27 m (*n* = 48)	Cuprolinic Blue/Nissl staining, Neurons/mm^2^,/ganglia, TH-IR, CGRP-IR	Neuron density: 38% loss of the SP neurons and 32% loss of the MP neurons at 27 m. Total neuron number: 24% loss of SP neurons and 31% MP neurons at 27 m. The density of the TH-IR swellings was a 3-fold increase by 16 m and a dramatic 12-fold increase by 24 m. The swollen CGRP-IR fibers occurred less frequently, were not found until 16 m, and were rare even at median age. (Correction factor for gut growth)	[29]
Sprague-Dawley ♂	Esophageal, pyloric and ileocecal sphincters (MP)	2–3 days, 6 weeks, 3 m, 25 m (*n* = 10 per age)	PGP9.5, VIP, CGRP, substance P, and dopamine-β-hydroxylase	All three regions: increase in the density of dopamine-β-hydroxylase- and substance P-IR nerve fibers. Decrease in density of CGRP-IR nerve fibers in the lower and ileocecal sphincters and VIP-IR nerve fibers in the pylorus.	[45]
**Mouse**					
NMRI/Bom ♂♀	Antrum, duodenum, colon (MP, SP)	1, 3, 12, 24 m (*n* = 36)	PGP9.5/ganglion	Loss of numbers of neurons per ganglion started at 12 m: MP: 40% loss in antrum, 60% loss in duodenum, 33% loss in colon. SP: 28% loss in antrum, 50% loss in duodenum, 40% loss in colon.	[46]
C57BL/6 ♂♀	Stomach, jejunum and colon (MP)	2, 12, 16, 20, 24 m (*n* = 60)	Interstitial cells of Cajal, PGP9.5, HuC/D, ChAT and NOS neurons	Reduction in total PGP9.5 and HuC/D protein at 20 m in colon. Decrease in ChAT-IR ganglia area and nerve fibers in the stomach from 16 m and starting at 20 m in the intestine. Reduction in NOS-IR neuron number per area in stomach between 12 m and 16 m and in the intestine at 20 m. Decreased interstitial cells of Cajal density over time from 16 m in stomach, 20 m in jejunum and 24 m in colon.	[20]
C57BL/6 ♂	Small intestine and colon (MP, SP, tunica mucosa)	6, 12, 18 m (*n* = 21)	βIII-tubulin, substance P, NOS GFAP-IR, and S100-IR (relative to βIII-tubulin-IR)	Decrease with age in the volume density of βIII-tubulin-IR at the MP and tunica mucosa of colon. No age-related differences in volume density of substance P-IR and GFAP-IR EGCs. Highest volume density of S100-IR at 18 m in both regions.	[47]
C57BL/6 ♂	Distal colon (MP)	3–4, 12–13, 18–19, 24–25 m (*n* = 18)	Total (HuC/D), calbindin, and nNOS neurons per area	No change in numbers of total neurons or subpopulations after correction but the density of MP neurons decreased between 3–4 m and 12–13 m. The density of nNOS-IR fibers in the MP increased remarkably with age, up to 18–19 m. Increased swollen processes of calbindin- and nNOS-IR neurons at 18–19 m and 24–25 m. (Correction for gut growth and stretch)	[26]
C57BL/6 ♂	Internal anal sphincter (circular muscle and mucosa)	3, 12–13, 18, 24–25 m (*n* = 12)	PGP9.5, nNOS, VIP, substance P, CGRP, and calretinin	No significant reduction in density of PGP9.5- and calretinin-IR neurons with age. Reduction in nNOS- and substance P-IR neurons with age in the circular muscle. Reduction in nNOS-, VIP-, and substance P-IR neurons in the anal mucosa with age. Increase in CGRP-neurons in both layers at 18 m.	[48]
C57BL/6 ♂♀	Colon (MP)	2, 5, 12 or 15 m (*n* = 4 per age)	Connexin-43 mRNA and staining, Western blot	Reduction in connexin-43-IR intensity within the MP and protein level in 15 m compared to 2 m but increasing in 5 m. Connexin-43 mRNA expression is double at 12 m compared to 2 m.	[49]
**Human**					
	Esophagus (MP)	20–40 y (*n* = 5), >70 y (*n* = 5)	Total (Geimsa stain)/area	22–62% loss	[24]
♂♀	Proximal duodenum (MP)	20–44, 45–64, 65–84 y (*n* = 30)	Silver nitrate and crystal-violet staining	Decreased by 16.26% and 16.46% in neural number in oldest group relative to middle-aged and youngest group, respectively.	[23]
♂♀	Jejunum (MP)	20–44, 45–64, 65–84 y (*n* = 30)	Silver nitrate and crystal-violet, H&E staining	Decrease in number of MP neurons of 25.93% in oldest compared to the youngest and of 23.32% in relation to the middle-aged.	[22]
♂♀	Ileum (MP)	42–71 y (*n* = 7), 78–86 y (*n* = 8)	PGP9.5, nitrergic (NADPH-d staining), calretinin neurons	Increased by 36% in nitrergic-IR and by 19% in calretinin-IR of PGP9.5-IR neurons in aged samples.	[50]
♂♀	Ileum, colon (MP)	10 days–92 y (*n* = 168)	Nitrergic (NADPH-d staining) /ganglion	Non-significant increase in proportion NADPH-d/ganglion with age.	[21]
♂♀	Small intestine (MP)	20–40, 69–76 y (*n* =12)	Giemsa staining	34% neuronal loss in the ganglia in whole small intestine, decreased by over 38% neuronal number in the duodenum.	[51]
♂♀	Colon (MP)	20–35 y (*n* = 6), >65 y (*n* = 6)	Total (Geimsa stain)/area	37% loss	[52]
♂♀	Colon (MP, SP)	Control 43–75 y (1♂, 9♀), STC patients 24–78 y (1♂, 25♀)	Neuron specific enolase, S100, CD34, and Bcl-2 immunostaining	Decreased significantly in neuron specific enolase-, S100-, Bcl-2-IR density in STC patients compared to controls in MP, SP but no differences in CD34-IR density. Increase significantly in the number of apoptotic enteric neurons in the MP of STC patients, whereas no differences in the SP.	[53]
♂♀	Colon (MP, SP)	33–99 y (9♂, 7♀)	PGP9.5, HuC/D, ChAT, and NOS neurons/ganglion and/mm length	No change in the total volume of PGP9.5-IR with age. Declined HuC/D- and ChAT-IR neuronal number in the MP but no change in the SP with increasing age. Increased nitrergic neurons in the MP but no change in the SP. 38% total loss in number of MP neurons between 30 and 60 y.	[54]
♂♀	Colonic muscle strips	37–65, 66–93 y (*n* = 22)	Gene expression of *GABAAR*	Decrease in the mRNA expression of the *GABAARα3* subunit. Increase in α2 and γ2 subunit but not statistical significance.	[55]
♂♀	Colon (MP)	4–12 m (3♂, 1♀), 48–58 y (1♂, 3♀), 70–95 y (4♂, 7♀)	Neural key genes, NADPH-d staining (NOS)	Decrease in relative gene expression of neural key genes, such as *NGFR*, *RET*, *NOS1,* and increase in *CHAT*. 16.6% loss of NOS-IR cell number in the aged donors compared to the babies. Regional differences in *RET*, *CHAT*, *TH*, and *S100B* gene expression in aged proximal and distal colon. No change in *SNCA*, *CASP3*, *CAT*, *SOD2*, and *TERT* expression. Decrease in gene expression of encoding sodium channel Nav1.1 and 1.5 with aging.	[56]
♂♀	Descending colon (MP, SP)	23–63 y (6♂, 7♀), elderly (66–81 y; 6♂, 4♀)	SOX-10, S100, and GFAP staining	Unaltered number of SOX-10-IR EGCs with age in MP and SP and no differences between adult males and females. Declined density in S100-IR EGCs among the elderly in the circular muscle and within the MP per ganglionic area. Little or no GFAP-IR EGCs in adult and elderly colon.	[57]

Abbreviations: MP, myenteric plexus; SP, submucosal plexus; E-d, embryonic day; P-d, postnatal day; m, months; y, years; ♂, male; ♀, female; IR, immunoreactivity; nNOS, neuronal nitric oxide synthase; PGP9.5, protein gene product 9.5; NADPH-d, nicotinamide adenine dinucleotide-diaphorase; ChAT, choline acetyltransferase; TH, tyrosine hydroxylase; CGRP, calcitonin gene-related peptide; VIP, vasoactive intestinal peptide; GFAP, glial fibrillary acidic protein; CD34, cluster of differentiation 34; Bcl-2, B-cell leukemia/lymphoma 2 protein; STC, slow transit constipation; GABAAR, γ-aminobutyric acid type A receptors; NGFR, nerve growth factor receptor; RET, ret proto-oncogene; S100, S100 calcium binding protein; SNCA, synuclein alpha; CASP3, caspase-3; CAT, catalase; SOD2, superoxide dismutase; TERT, telomerase reverse transcriptase; SOX-10, SRY-box transcription factor 10; EGCs, enteric glial cells.

## Data Availability

Not applicable.

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
