# Peer review of "The Aging Enteric Nervous System"

_ijms, 2023, doi:10.3390/ijms24119471_

Round 1
Reviewer 1 Report
Overall, this is a well written and comprehensive review publication and offers a good overview on the role of age for functioning of enteric nervous system. However, before publication some points need to be clarified.
Line 33 – ENS is not a part of ANS. Nowadays it is considered as an independent structure.
Line 40 – Both Aurebach and Meissner plexus names are no longer used. Please delete them.
Line 45 –this is not true. Submucous plexus is not present in the stomach. There are submucous ganglia but they do not form any plexus.
Table 1 – The authors should ensure that they use term “expression” in relation to genes only
Line 461 – Not only people change their eating behavior. Also, animals do.
Line 463 - Some plants-derived nutritional factors (like for example lectins isolated from red-kidney bean) also influence the structure of GIT organs of young animals including changes in enteric neurons profile (see Kruszewska D, et al., Enteral crude red kidney bean (Phaseolus vulgaris) lectin--phytohemagglutinin-induces maturational changes in the enterocyte membrane proteins of suckling rats. Biol Neonate. 2003;84(2):152-8. and Zacharko-Siembida A et al. Immunolocalization of NOS, VIP, galanin and SP in the small intestine of suckling pigs treated with red kidney bean (Phaseolus vulgaris) lectin. Acta Histochem. 2013 Apr;115(3):219-25. In my opinion this is in line with current review and the authors should briefly address this issue as well as acknowledge these works.
Author Response
We appreciate the kind and helpful advice we received by the reviewers to improve the current revised version of our manuscript. We have addressed all of the questions and advice and corrected the manuscript accordingly based on the comments made by the reviewers. Furthermore, we have highlighted all changes and corrections in yellow, in order to make it easier for identifying the changes. A point-by-point list of responses and explanations is provided below.
Response to Reviewer 1 Comments
Overall, this is a well written and comprehensive review publication and offers a good overview on the role of age for functioning of enteric nervous system. However, before publication some points need to be clarified.
We are thankful to the reviewer for reading our manuscript thoroughly and taking the time for providing his/her valuable comments in order to improve the manuscript.
Point 1: Line 33 – ENS is not a part of ANS. Nowadays it is considered as an independent structure.
Response 1: We agree with your comment and we changed in the revised manuscript as follows:
Line 33: The ENS is the largest and most intricate division of the peripheral nervous system, extending from the upper esophagus to the internal anal sphincter and connect to the liver, gall bladder, biliary tract, and pancreas [2].
Point 2: Line 40 – Both Aurebach and Meissner plexus names are no longer used. Please delete them.
Response 2: We thank you for pointing out this. We deleted Aurebach and Meissner plexus names as in line 40.
Point 3: Line 45 –this is not true. Submucous plexus is not present in the stomach. There are submucous ganglia but they do not form any plexus.
Response 3: We referred this information from reference Kulkarni, S.; Ganz, J.; Bayrer, J.; Becker, L.; Bogunovic, M.; Rao, M. Advances in Enteric Neurobiology: The "Brain" in the Gut in Health and Disease. J Neurosci 2018, 38, 9346-9354, doi:10.1523/jneurosci.1663-18.2018 but we agree that this information is not correct. Now we referred to the reference Tobias, A.; Sadiq, N.M. Physiology, Gastrointestinal Nervous Control. In StatPearls; Treasure Island (FL), 2022. We corrected this information in lines 43-45 : The submucosal plexus lies beneath the mucosa, closer to the intestinal lumen, and in the small and large intestines, but not in the stomach or esophagus [2].
Point 4: Table 1 – The authors should ensure that they use term “expression” in relation to genes only
Response 4: We are sorry for this. We corrected these mistakes accordingly:
Significant decline in nNOS mRNA, protein level and nNOS-IR nerve fiber with age.
Decreased significantly neuron specific enolase-, S100-, Bcl-2-IR density in STC patients compared to controls in MP, SP but no differences in CD34-IR density.
Point 5: Line 461 – Not only people change their eating behavior. Also, animals do.
Response 5: We agree with the reviewer comment and add a respective half sentence and two respective references (line 470).
Point 6: Line 463 - Some plants-derived nutritional factors (like for example lectins isolated from red-kidney bean) also influence the structure of GIT organs of young animals including changes in enteric neurons profile (see Kruszewska D, et al., Enteral crude red kidney bean (Phaseolus vulgaris) lectin--phytohemagglutinin-induces maturational changes in the enterocyte membrane proteins of suckling rats. Biol Neonate. 2003;84(2):152-8. and Zacharko-Siembida A et al. Immunolocalization of NOS, VIP, galanin and SP in the small intestine of suckling pigs treated with red kidney bean (Phaseolus vulgaris) lectin. Acta Histochem. 2013 Apr;115(3):219-25. In my opinion this is in line with current review and the authors should briefly address this issue as well as acknowledge these works.
Response 6: We thank the review for his/her suggestion regarding two references: Kruszewska D, Kiela P., Ljungh A., Erlwanger K.H., Weström B.R., Linderoth A., Pierzynowski S.G. Enteral crude red kidney bean (Phaseolus vulgaris) lectin-phytohemagglutinin-induces maturational changes in the enterocyte membrane proteins of suckling rats. Biol Neonate. 2003;84(2):152-8. and Zacharko-Siembida A., Piedra J.L.V, Szymańczyk S., Arciszewski M.B. et al. Immunolocalization of NOS, VIP, galanin and SP in the small intestine of suckling pigs treated with red kidney bean (Phaseolus vulgaris) lectin. Acta Histochem. 2013 Apr;115(3):219-25.
Following the reviewer’s suggestion, we checked the literature and found only two more references that studied red-kidney bean Phaseolus vulgaris lectin. Firstly, another paper of Zacharko-Siembida et al. assessed changes in expression of calbindin 28 kDa after Phaseolus vulgaris lectin administration in suckling piglets (Zacharko-Siembida A., Piedra J.L.V., Arciszewski M.B. Changes in expression of calbindin 28 kDa in the small intestine of red kidney bean (Phaseolus vulgaris) lectin-treated suckling piglets. Polish Journal of Veterinary Sciences Vol. 16, No. 2 (2013), 201–209). However, these three references investigated impact of plants-derived nutritional factors (lectins isolated from red-kidney bean) on pre-weaning animals and not on aging, which are not in line to our review’s concept regarding aging ENS.
Secondly, Zheng et al. showed that Phaseolus vulgaris lectin transported from the gut to the GFP-labeled dopaminergic neurons of Caenorhabditis elegans and reduced the number and fluorescent intensity of GFP-labeled dopaminergic neurons, suggesting its toxic activity to dopaminergic neurons as an etiologic agent for Parkinson’s disease (Zheng J., Wang M., Wei W., Keller J.N., Adhikari B., King J.F., King M.L., Peng N. and Laine R.A. (2016) Dietary Plant Lectins Appear to Be Transported from the Gut to Gain Access to and Alter Dopaminergic Neurons of Caenorhabditis elegans, a Potential Etiology of Parkinson’s Disease. Front. Nutr. 3:7. doi: 10.3389/fnut.2016.00007). However, this study reports on C. elegans which does not really fit our review as we mostly focussed on mice, rats, and guinea-pigs. Therefore, we consider not to include the studies of red-kidney bean Phaseolus vulgaris lectin.

Reviewer 2 Report
The authors of this well-prepared narrative review have aimed to describe the ageing enteric nervous system using studies describing these changes in animal models and humans alike. The review is comprehensive and flows logically into each section. The results of the studies are well described in both the tables and figures. Well done. The only comments here, which are relatively minor, include:
There is no discussion or mention of the splanchnic nerves/divisions. These are intimately related to the ganglia of the colon, as well as inferior hypogastric plexus. Since the Vagus nerve is mentioned, providing a brief summary of the function of the splanchnic nerve would make this review more complete. However, this is more a recommendation than anything.
There is a sentence listed in the text: "Differential expression in a sex-dependent manner was reported in both mice and humans such as expression of the Y-linked gene (DEAD-Box Helicase 3 Y-Linked, Ddx3y) only in males and expression of X-inactive specific transcript (Xist) only in females [10]." It is not clear what the authors are aiming to demonstrate here or what impact this has on the ENS. Could the authors be clarify in more detail and expand on this in this paragraph or, preferably, remove this as these are specifically related to sexual development and do not fit into this narrative.
Section 4 does not provide an extensive summary of how the ageing ENS can induce PD or AD, rather it provides a brief overview of studies exploring this novel area of research. Please adjust the aim of this section, which currently reads "Therefore, here we aim at pinpointing at relationship between the ENS and CNS in the most popular neurodegenerative diseases of the elderly—Parkinson's disease and Alzheimer's disease." The aim here is not really to 'pinpoint' the relationship rather to provide a potential mechanism by which changes in the ageing ENS could predispose to the development of PD or AD or partially contribute to their development. This is undoubtedly multi-faceted and this is not comprehensively demonstrated in this portion of the review. Please adjust the heading include the words 'the potential' in front of Disease-related pathology of the ENS to reflect this as well. However, it should be noted what is presented in this section is in keeping with what is presented in the rest of the article - well done.
Please double-check and proofread the paper to ensure that all abbreviations are described on their first use. Many are not described or only used once, which, in this, they should not be used as abbreviations and simply written in full. This will improve clarity for the reader. Further there are words missing throughout the course of the article include of, the, is from, etc. There are also some plural forms of a word used when they should not be and vice versa (e.g. humans instead of human in the 2nd paragraph). Some words are used in incorrectly (e.g. here we aim at pinpointing at relationship - should replace the second at with the word the). There are inconsistencies in where hypens are applied and when they are not and the use of abbreviations in brackets following the full definition is inconsistent. Finally, the authors appear to switch between UK and US English (e.g. estrogens = US, but faecal = UK) - this just needs to be consistent. The reviewer appreciates that this is probably the authors second language and the use of certain, abbreviations, punctuation, etc. can be challenging, but some basic editing assistance here would greatly enhance the readability of this well prepared document (i.e. it is mostly well-written, but a good thorough proofread will just improve this that little bit more).
Author Response
We appreciate the kind and helpful advice we received by the reviewers to improve the current revised version of our manuscript. We have addressed all of the questions and advice and corrected the manuscript accordingly based on the comments made by the reviewers. Furthermore, we have highlighted all changes and corrections in yellow, in order to make it easier for identifying the changes. A point-by-point list of responses and explanations is provided below.
Response to Reviewer 2 Comments
The authors of this well-prepared narrative review have aimed to describe the ageing enteric nervous system using studies describing these changes in animal models and humans alike. The review is comprehensive and flows logically into each section. The results of the studies are well described in both the tables and figures. Well done.
We appreciate the kind comments of the reviewer, and we are grateful for the suggestions for improving quality of our manuscript.
Point 1: There is no discussion or mention of the splanchnic nerves/divisions. These are intimately related to the ganglia of the colon, as well as inferior hypogastric plexus. Since the Vagus nerve is mentioned, providing a brief summary of the function of the splanchnic nerve would make this review more complete. However, this is more a recommendation than anything.
Response 1: Following the reviewer’s recommendation, we added two sentences regarding function of two innervation systems accordingly:
Line 46: The ENS has a bidirectional connection to the CNS via the vagus nerve, pelvic nerves, splanchnic nerves and sympathetic pathways, including afferent (sensory) innervation and efferent (motor) innervation [5,6].
Line 52: Vagal innervation conveys mechanoreceptive and chemoceptive information from the esophagus, stomach, and intestine to the CNS, but not pain signals, and regulates gastric propulsion and motility, gastric acid production and hormone release in the ENS [5]. Unlike the vagal nerve, the splanchnic nerves carry pain conducting visceral afferent fibers and visceral efferent fibers, which inhibit GI motility and secretion, stimulate gluconeogenesis, glycogenolysis and glucose release, secretion of catecholamines by chromaffin cells, and allow for transmission of the intractable visceral pain [6].
Point 2: There is a sentence listed in the text: "Differential expression in a sex-dependent manner was reported in both mice and humans such as expression of the Y-linked gene (DEAD-Box Helicase 3 Y-Linked, Ddx3y) only in males and expression of X-inactive specific transcript (Xist) only in females [10]." It is not clear what the authors are aiming to demonstrate here or what impact this has on the ENS. Could the authors be clarify in more detail and expand on this in this paragraph or, preferably, remove this as these are specifically related to sexual development and do not fit into this narrative.
Response 2: We aim to give some examples of the shared program and differences in the ENS of mice and humans. This sentence ‘’Differential expression in a sex-dependent manner was reported in both mice and humans such as expression of the Y-linked gene (DEAD-Box Helicase 3 Y-Linked, Ddx3y) only in males and expression of X-inactive specific transcript (Xist) only in females [9]’’ is therefore the second example for a shared program between mouse and man, regarding genes that are expressed in a sex-dependent manner. These genes are not ENS specific but have been found expressed in the ENS.
Point 3: Section 4 does not provide an extensive summary of how the ageing ENS can induce PD or AD, rather it provides a brief overview of studies exploring this novel area of research. Please adjust the aim of this section, which currently reads "Therefore, here we aim at pinpointing at relationship between the ENS and CNS in the most popular neurodegenerative diseases of the elderly—Parkinson's disease and Alzheimer's disease." The aim here is not really to 'pinpoint' the relationship rather to provide a potential mechanism by which changes in the ageing ENS could predispose to the development of PD or AD or partially contribute to their development. This is undoubtedly multi-faceted and this is not comprehensively demonstrated in this portion of the review. Please adjust the heading include the words 'the potential' in front of Disease-related pathology of the ENS to reflect this as well. However, it should be noted what is presented in this section is in keeping with what is presented in the rest of the article - well done.
Response 3: We agree with the reviewer suggesstion. We changed the section 4 accordingly to the reviewer’s suggestion:
Line 504: 4. Potential disease-related pathology of the ENS
Line 511: Therefore, here we aim at pinpointing the potential mechanism by which changes in the aging ENS could be predisposing to the development of the popular neurodegenerative diseases of the elderly—Parkinson's disease and Alzheimer's disease.
Point 4: Comments on the Quality of English Language: Please double-check and proofread the paper to ensure that all abbreviations are described on their first use. Many are not described or only used once, which, in this, they should not be used as abbreviations and simply written in full. This will improve clarity for the reader. Further there are words missing throughout the course of the article include of, the, is from, etc. There are also some plural forms of a word used when they should not be and vice versa (e.g. humans instead of human in the 2nd paragraph). Some words are used in incorrectly (e.g. here we aim at pinpointing at relationship - should replace the second at with the word the). There are inconsistencies in where hypens are applied and when they are not and the use of abbreviations in brackets following the full definition is inconsistent. Finally, the authors appear to switch between UK and US English (e.g. estrogens = US, but faecal = UK) - this just needs to be consistent. The reviewer appreciates that this is probably the authors second language and the use of certain, abbreviations, punctuation, etc. can be challenging, but some basic editing assistance here would greatly enhance the readability of this well prepared document (i.e. it is mostly well-written, but a good thorough proofread will just improve this that little bit more).
Response 4: We are sorry for having overseen some language mistakes in the review. We asked a native English speaker to proofread thoroughly our review and now we corrected mistakes accordingly to the reviewer’s and native English speaker’comments to have consistent usage of US English language. Examples are such as fecal (line 510), esophagus (Table 1, line 34, 147, 173), signaling (line 318, 368, 392, 449, 459), fiber (line 124, 136, 154, 156, 158, 531, and Table 1), etc. Many other changes are highlighted in yellow throughout the revised review, we felt that inserting them in this reply letter would be superfluous.
Besides, we now described all abbreviations on their first use in the revised manuscript. Examples are such as S- (synaptic or Dogiel Types I) and AH (after-hyperpolarization or Dogiel Types II)-type submucosal neurons (line 378).
Some abbreviations used once are now written in full, for example, dopamine-β-hydroxylase (Table 1), tnterstitial cells of Cajal (Table 1), colony stimulatory factor 1 (line 395), etc.
The reviewer comments that there are inconsistencies in where hypens are applied. We are not sure what the reviewer means. We used different hyphens for different purposes such as: hyphen (-) as a punctuation mark used to link multiple words; the en dash (–) are used in number ranges; while the em dash (—) are used to break up a sentence. We would like to ask the reviewer to check our revised manuscript and if the reviewer find the hyphen types that are used incorrectly, please point them out, then we will correct them.
